# Impact of Transgenic Maize Ruifeng125 on Diversity and Dynamics of Bacterial Community in Rhizosphere Soil

**DOI:** 10.3390/microorganisms12091763

**Published:** 2024-08-25

**Authors:** Chaofeng Hao, Xinyao Xia, Chao Xu, Hongwei Sun, Fan Li, Shuke Yang, Xiaohui Xu, Xingbo Lu

**Affiliations:** 1Shandong Key Laboratory for Green Prevention and Control of Agricultural Pests, Institute of Plant Protection, Shandong Academy of Agricultural Sciences, Jinan 250100, China; chaofenghao2015@163.com (C.H.); bioxxy@163.com (X.X.); hongweisun@126.com (H.S.); lfzjnd@163.com (F.L.); yangshuke426@126.com (S.Y.); 2Key Laboratory for Safety Assessment (Environment) of Agricultural Genetically Modified Organisms, Ministry of Agriculture and Rural Affairs, Jinan 250100, China; 3Institute of Germplasm Resources and Biotechnology, Tianjin Academy of Agricultural Sciences, Tianjin 300381, China; xuchao5131@163.com

**Keywords:** rhizosphere soil, bacterial community, infusion Bt protein, transgenic maize

## Abstract

With the development of commercialized planting of genetically modified crops, their ecological security risks remain a key topic of public concern. Insect-resistant genetically modified maize, Ruifeng125, which expresses a fusion Bt protein (Cry1Ab-Cry2Aj), has obtained the application safety certificate issued by the Chinese government. To determine the effects of Ruifeng125 on the diversity and dynamics of bacterial communities, the accumulation and degradation pattern of the fusion Bt protein in the rhizosphere soil of transgenic maize were detected. Results showed that the contents of Bt protein varied significantly at different developmental stages, but after straw was returned to the field, over 97% of Bt proteins were degraded quickly at the early stages (≤10 d) and then they were degraded at a relatively slow rate. In addition, the variations in bacterial community diversity in the rhizosphere soil were detected by 16S ribosomal RNA (Rrna) high-throughput sequencing technology. A total of 44 phyla, 435 families, and 842 genera were obtained by 16S rRNA sequencing, among which *Proteobacteria*, *Actinobacia*, *Acidobacter Acidobacterium*, and *Chloroflexi* were the dominant taxa. At the same developmental stage, no significant differences in soil bacterial diversity were detected between Ruifeng125 and its non-transgenic control variety. Further analysis revealed that developmental stage, rather than the transgenic event, made the greatest contribution to the changes in soil microbial diversity. This research provides important information for evaluating the impacts of Bt crops on the soil microbiome and establishes a theoretical foundation for their environmental safety assessment.

## 1. Introduction

Genetically modified crops (GMCs) are created by the introduction of artificially isolated and modified exogenous genes into the genome of crops through genetic engineering technology. The exogenous genes can be stable and passed on to the next generation of crops [1]. Generally, the original traits of transgenic crops will be replaced by new stability traits, such as herbicide resistance, insect resistance, and disease resistance [2]. *Bacillus thuringiensis* (Bt) toxin proteins are a class of insecticidal crystal proteins derived from the Gram-positive bacterium *B. huringiensis* [3,4,5]. These proteins are broken down into small active toxin fragments in the target organism and can bind to the organism’s intestinal epithelial cells and cause perforations, thereby affecting the osmotic balance, and eventually resulting in the death of the target organism [6,7,8]. The plant–microorganism interaction is a complex dynamic process, which includes both beneficial symbiotic complementation [9] and the invasion of pathogenic microorganisms [10]. Microorganisms play a key role in plant growth, stress resistance, nutrition, productivity, and product quality [11,12]. Soil microbial biomass and its turnover, as a reserve of effective nutrients for plants, play an important role in the transformation and circulation of carbon, nitrogen, phosphorus, and sulfur in soil [13,14]. The soil ecosystem is the place where plants and soil microorganisms conduct material circulation, energy transfer, and information exchange [15]. The exogenous gene expression products of transgenic plants will enter the soil ecosystem through various ways [16], which may affect the biological structure of the soil ecosystem.

Whether the planting of transgenic crops will affect soil microorganisms is one of the focuses of relevant studies. The conclusions drawn by previous studies in this area are not consistent, and even contradictory. For example, it was found that the colonization of arbuscular mycorrhizal fungi in the roots of transgenic maize Bt176 was significantly reduced [17]. Compared with the control lines, the planting of transgenic maize MON810 had no significant impact on the fungi community in rhizosphere soil during the whole growth stages. However, the bacterial and actinomycete communities in the rhizosphere soil of the two maize varieties were significantly different at the silking stage and seedling stage, respectively [18]. Meanwhile, other research results revealed that the planting of transgenic maize had no significant impact on soil microorganisms. For example, Andreas et al. showed that the Bt protein secreted by Bt transgenic maize during its growth period had no effect on soil microorganisms [19]. Zou found that the one-year planting of transgenic maize had no significant effect on the bacterial community structure of maize rhizosphere soil [20].

More and more scientists believe that scientific evaluation of whether genetically modified crops have a significant impact on soil microbial diversity should follow the case-by-case evaluation principle [21]. Combined with the properties and stability of exogenous proteins, as well as their accumulation and degradation patterns in the soil, we could systematically analyze the changes in soil microbial diversity at different stages and determine the impact of specific genetically modified varieties on soil microbial diversity.

The insect-resistant genetically modified maize Ruifeng125, with a fusion Bt protein (Cry1Ab-Cry2Aj), is one of the first transgenic maize variants that has obtained the application safety certificate issued by the Chinese government [22]. It has excellent insecticidal activities against Lepidoptera pests, such as Asian corn borer, oriental armyworms, and bollworm. With the acceleration of China’s pilot promotion of genetically modified crops, it is necessary to explore the impact of Ruifeng125 on the dynamics and diversity of microbial communities in the rhizosphere soil. In this study, we first analyzed the accumulation and degradation dynamics of the contents of the fusion Bt protein in the rhizosphere soils of Ruifeng125 and its control variety. Then, the abundance and diversity of bacterial communities in the rhizosphere soil of the two maize varieties were analyzed by 16S rRNA sequencing. The results provide valuable information for people to understand the effects of the transgenic maize Ruifeng125 on the soil ecosystem and guarantees for the sustainable development of modern agriculture in China.

## 2. Materials and Methods

### 2.1. Plant Materials

The insect-resistant transgenic maize Ruifeng125 and its non-transgenic control Xianyu 335 were provided by the Institute of Plant Protection, Shandong Academy of Agricultural Sciences.

### 2.2. Test Design

Both the transgenic maize Ruifeng125 and its non-transgenic control were planted in PVC plastic pots (80 cm in diameter and 50 cm in height), which were filled with 25 kg of soil collected from the same field. Each of the maize varieties was planted in 30 pots, with 5 pots per treatment. Before sowing, each pot was given the same amount of sterile water. Then, 10 seeds were buried in each pot, and the number of plants was fixed to 3 plants/pot at the V3 stage. During the whole growth period, no pesticide was used to control diseases or pests.

### 2.3. Collection of Rhizosphere Soil

The rhizosphere soil samples of the two maize varieties were collected using the shake method [23] at the seedling stage, jointing stage, botting stage, tasseling stage, milk ripening stage, and full-ripe stage. Soils collected from three plants in one pot were firstly filtered with 40-mesh sieves to remove impurities, and the pure soils were collected into a Ziplock bag as one biological replicate. All the filtered soil samples were put into dry ice and transferred to a refrigerator at −80 °C before sequencing.

### 2.4. DNA Extraction and PCR Amplification

Microbial DNA from soil samples was extracted using Fast DNA@SPIN Kit for Soil kit (116560200). A NanoDrop 2000C (Thermo Fisher Scientific, Waltham, MA, USA) micro spectrophotometer was used to detect the concentration and quality of extracted DNA (OD260/280), and 1.2% agarose gel electrophoresis was used to detect the quality of DNA. 

Using the purified DNA samples as templates, the V3-V4 region of 16S rRNA gene universal primers (338F 5′-ACTCCTACGGAGGCAGCAG-3′, 806R 5′-GACTACHVGGGTWTCTAAT-3′) [24] were used for PCR amplification. The amplification procedure was as follows: 95 °C pre-denaturation for 3 min, 35 cycles (95 °C denaturation for 30 s, 55 °C annealing for 30 s, 72 °C extension for 30 s), and finally 72 °C extension for 10 min. The amplification system was 20 μL, 4 μL 5 × FastPfu buffer, 2 μL 2.5 mM dNTPs, 0.8 μL primer (5 μM), 0.4 μL FastPfu polymerase; 10 ng DNA template.

### 2.5. Detection of Bt Protein Contents in the Rhizosphere Soil

The Envirologix CryAb/CrylAc plate kit was used to determine the Bt protein contents in the soil samples [25]. The standard curves were drawn according to the manufacturer’s instructions with six concentrations of positive standard samples (1 ng/mL, 0.8 ng/mL, 0.5 ng/mL, 0.4 ng/mL, 0.25 ng/mL, and 0.125 ng/mL, CryAb/CrylAc). Then, the Bt protein content of the soil sample was calculated according to the standard curve.

### 2.6. Illumina Miseq Sequencing and Data Processing

According to the standard operating procedures of the Illumina MiSeq platform (Illumina, San Diego, CA, USA), the purified amplified fragment was constructed into PE 2 × 300 library, using the MiSeq PE300 platform for sequencing (Illumina, San Diego, CA, USA.). The original FASTQ file uses Trimmatic (version 0.39) software for quality control and is spliced by FLASH software according to the following standards: (1) Set a 50 bp window. If the average quality value in the window is less than 20, cut off all the sequences at the back end of the base from the front end of the window, and then remove the sequences with length less than 50 bp after quality control. (2) The sequences at both ends are spliced according to the overlapping base overlay. The maximum mismatch rate between overlays is 0.2, and the length should be greater than 10 bp. Remove the sequence that cannot be spliced. (3) The sequence is divided into each sample according to the barcode at the beginning and end of the sequence and the primer. The barcode needs to be matched accurately. The primer allows the mismatch of two bases and removes the sequence with fuzzy bases.

UPARSE software using (version 7.1) OTU (Operational Taxonomic Units) clustering was performed on the sequence according to 97% similarity, and single sequence and chimera were removed in the process of clustering. An RDP classifier was utilized (http://rdp.cme.msu.edu/). Each 16S rRNA sequence was annotated for species classification, compared with the Silva database (Silva138), and the matching threshold was set at 70%. The data analysis was conducted on the Meggitt Shengxin cloud platform (www.i-sanger.com). The Wilcox rank-sum test was used to analyze the significance difference between paired groups, and the method of FDR was used to test and correct the *p* value. At the OTU level, the threshold FDR < 0.01 and the *p* value < 0.05 determine that there is a significant difference in the relative abundance of bacteria between the two groups of samples. The obtained 16S sequence was compared with ClusterW to clarify the evolution relationship of OTU in the sample and the comparison results are shown in the form of a circular phylogenetic tree with 1000 times bootstrapping using MEGA v5.05. Linear discriminant analysis effect size (LEfSe) was determined using the R package microeco.

## 3. Results

### 3.1. The Accumulation and Degradation Dynamics of Bt Protein in the Rhizosphere Soil

The content of Bt protein in the rhizosphere soil of the transgenic maize Ruifeng125 changed greatly during the vegetative growth periods. From the perspective of the whole growth cycle of maize, the content of Bt protein reached the maximum value at the tasseling stage, reaching 13.4 ng/g, and then the content of Bt protein in the rhizosphere soil at the following stages gradually decreased (Figure 1).

At the early stage after corn straw was returned to the field, the Bt protein in the soil degraded rapidly. Six days after returning the corn straw to the field, over 90% of the Bt protein was degraded, and the content of Bt protein decreased from 937.60 ng/g to 89.82 ng/g. From the 7th day, the degradation rate of Bt protein slowed down, and by the 10th day, the degradation rate of Bt protein reached 97.26%. At the 20th, 40th, 70th, 110th, and 160th days, the degradation rate was even more slow. At the 160th day, the degradation rate of the sample Bt protein was 99.97%. On the 170th day, the content of Bt protein in the sample was lower than the detection limit of the kit, and no active component of Bt protein was detected (Table 1).

### 3.2. Statistics and Analysis of Sequencing Data

The V3-V4 region of the 16S rRNA gene of bacteria of rhizosphere soil samples was sequenced on the Illumina Miseq PE 300 (Illumina, San Diego, CA, USA) platform, generating 1,382,450 high-quality reads and 606,846,719 base numbers, with the 99.99% average inter-group sequencing coverage, indicating that the data measured in this experiment can truly reflect the composition of most bacterial groups in the sample (Table 2). The curve of the number of observed OTUs tended to be stable (Figure 2), which indicated that the amount of sequencing data of the sample was enough to cover most microbial taxa.

### 3.3. Changes of Bacterial Community Structure in Rhizosphere Soils of Transgenic Maize Ruifeng125 and Its Control Line

A total of 5466 OTUs were obtained in transgenic maize Ruifeng125 rhizosphere soil samples, belonging to 44 phyla, 107 classes, 216 orders, 427 families, 818 genera, and 1712 species, of which 29 species were not detected in the rhizosphere soil of the non-transgenic control. The bacterial diversity in the rhizosphere soil samples of the conventional control was slightly higher, with 5540 OTUs in total, belonging to 44 phyla, 107 classes, 216 orders, 426 families, 828 genera, and 1732 species, and 49 species of bacteria were unique to them (Figure 3).

The corresponding sequence of the top 100 OTUs with the most abundance among all the samples was selected to construct an evolutionary tree according to the NJ (neighbor joining) adjacency method, presenting the species composition relationship of bacteria in the rhizosphere soil in the form of a circular evolutionary tree. These bacteria belong to nine phyla, including Proteobacter, Actinobacter, Chloroflexi, Firmicutes, Acidobacter, Nitrospirae, and Germatimonadetes. In the ring species evolution tree, the number of OTUs in three phyla is ≥10%. Proteobacteria is the largest group at the phyla level, including 33 OTUs, followed by Actinobacteria (24 OTUs), Chloroflexi (11 OTUs), and Firmicutes (10 OTUs). OTU1289 (Bacillus niacini) owned 10,550 reads, and its abundance was significantly higher than that of other OTUs (Figure 4).

### 3.4. Beta Diversity of Rhizosphere Bacterial Community of the Two Maize Lines

To further analyze the impacts of transgenic maize Ruifeng125 on bacterial communities in rhizosphere soil, principal coordinate analysis (PCoA) analysis of Bray–Curtis distances (PERMANOVA with Adonis test) was conducted. Firstly, using developmental stages as variables, we performed PCoA analyses on transgenic maize Ruifeng125 and its control (Figure 5A,B). The results showed that there were significant differences in the bacteria communities at different development stages in both transgenic (Figure 5A, R^2^ = 0.389, *p* = 0.003) and control maize (Figure 5B, R^2^ = 0.425, *p* = 0.001). In detail, the tasseling stage acted as a dividing line that distinguishes the bacterial communities into two parts in the two maize varieties, with the early stages occupying the position on one side, and the later stages occupying the position on the other side (Figure 5A,B). Considering that the tasseling stage is a critical turning point from nutritional growth to reproductive growth, the changes in microbial diversity in the two maize varieties may be a result of developmental stage changes. However, when the difference in maize varieties was taken as the only variable, no significant difference was observed in the bacterial communities among the seeding stage (*p* = 0.7), jointing stage (*p* = 0.3), botting stage (*p* = 0.3), tasseling stage (*p* = 0.9), milk ripening stage (*p* = 0.1), and full-ripe stage (*p* = 0.3) (Figure 5C–H). Therefore, we suggest that the main factor for the changes in the diversity of rhizosphere bacterial communities is the growth stage rather than the variety.

### 3.5. Difference in Bacterial Community Abundance in Two Rhizosphere Soils

After calculating the mean value of two kinds of corn rhizosphere soil samples at different developmental stages, the abundance differences of symbiotic bacteria between them were compared at the genus level. On the whole, the dominant bacterial communities in the rhizosphere soil of the two maize varieties at the same growth stages have strong consistency. The dominant bacteria include acidobacteria (norank_c_Acidobacteria), Bacillus (Bacillus), norank_f_Anaerolineaceae, nitrifying spirochetes (Nitrospira), and nitrogen-fixing bacteria (Nitrospira) (Figure 6A). However, the abundance of the dominant bacteria in the rhizosphere soil of the two maize varieties at different growth stages is different (Figure 6B). For example, the abundance of acid bacilli (norank_c_Acidobacteria) in the rhizosphere soil of transgenic maize is higher than that of the control at the first four growth stages, but lower than that of the control at the milk ripening and full-ripe stages. The abundance of some bacteria groups only showed significant differences in the rhizosphere soil of a single stage, such as OTU belonging to Gemmatimonadaceaein in the rhizosphere soil of transgenic maize at the milky stage. The abundance of Gaiellares was significantly higher than that of the control in the same period. At the tasseling stage, the abundance of Azotobacter in the rhizosphere soil of transgenic maize was significantly lower than that of the control. In the whole growth process of the two kinds of maize, only two of the ten bacterial communities with the highest abundance have significant differences (Figure 6C).

### 3.6. Biomarkers between Transgenic Maize Ruifeng125 and Non-Transgenic Control along the Six Developmental Stages

Linear discriminant analysis (LDR > 3) was then conducted to investigate the genus-level biomarkers between the control group and the transgenic group at six growth stages. As shown in Figure 7A, at the seedling stage, the biological marker genera of the control group (D) were *Rhodococcus*, while the biological marker genera of transgenic maize Ruifeng125 (Z) were *Gitt_GS_136*. At the jointing stage, it was found that *Hinschia*, *B1_7BS*, and *Pseudolabrys* were enriched in the control group, while *Bacillus* was enriched in transgenic maize Ruifeng125 (Figure 7B). *ABS_19*, *Chryseolinea*, *Bradyrhizobium*, and *Steroidobacter* were most abundant in Xianyu335, while *Hydrogenophaga*, *RB41*, *Roseateles*, and *Azohydrononas* were the most abundant bacteria in transgenic maize Ruifeng125 at the botting stage (Figure 7C). At the tasseling stage, we found *Blautia* was enriched in Xianyu335, while *Gaiellales* and *Nitrospira* were enriched in Ruifeng125 (Figure 7D). Overall, increasing bacterial taxa were identified as biomarkers across the maize developmental stages, indicating that the assembly of microbial communities was a dynamic process, and the late differentiation was gradually obvious.

## 4. Discussion

In soil, microorganisms are not distributed evenly. The closer they are to the roots of plants, the more types and quantities of microorganisms there are, and the stronger their metabolic activity. The number, types, and metabolic activities of rhizosphere soil microorganisms are several times or even dozens of times greater than those in the periphery. The number of genes possessed by rhizosphere microorganisms far exceeds the number of genes in plants. The efficacy of rhizosphere microorganisms in improving plant nutrition, preventing soil borne diseases, resisting acidic soil, and remediation of heavy metal and organic pollution in soil has been uncovered in the past 30 years [26]. Thus, the diversity of rhizosphere soil microbial communities can serve as an important indicator for soil environmental changes.

The debate over the environmental impacts of genetically modified insect-resistant crops has lasted over twenty years. Bt crops can produce Bt toxins in the plants, which can efficiently kill the target pest throughout the plant’s whole growth periods. Bt proteins can enter the soil ecosystem mainly through three ways: root exudates, pollen grains, and residues. Once they enter the soil, they form complexes with other active particles. The stability of Bt proteins is affected by various soil conditions, including soil type, pH, temperature, and water content [27,28,29]. Previous studies found that most of the Bt proteins in the soil were degraded rapidly at the early stage, and trace amounts of Bt proteins degraded stably at the later stage [30,31,32]. Consistent with other previous studies [33], the contents of the Bt protein in the rhizosphere soil of transgenic maize Ruifeng125 changed along with the plant developmental process (Figure 1). The content of Bt protein reached the maximum at the tasseling stage and the minimum at the jointing stage. In addition, the degradation patterns of Bt protein in the soil after returning straw to the field also showed a similar tendency with other researchers [34]. It was found that after returning straw to the field, Bt protein secreted by transgenic maize Ruifeng125 degraded rapidly and significantly in the soil in the first six days (over 90%), and then the degradation rate gradually slowed down (Table 1). At the 170th day, we could not detect the expression of Bt protein in the soil (Table 1). The results suggest that the tiny amounts of Bt protein secreted by Ruifeng125 will be gradually degraded with the decomposition of plant residues and will not remain stable in the soil for a long time

The rhizosphere soils of transgenic maize Ruifeng125 and its non-transgenic control Xianyu 335 have roughly the same dominant bacterial communities in each growth period (Figure 6). During the whole growth process, the dominant flora with a significant difference in community abundance in the rhizosphere soil of the two maize species is only MSB-1E8 (Figure 6). In addition, there are seven bacteria with significant differences in the tasseling stage, milk ripening stage, and full-ripe stage (Figure 6). With the exception of Anaerolineaceae, the community abundance of species in the rhizosphere soil of transgenic maize Ruifeng125 is significantly higher than that of the non-transgenic control group. The diversity of bacterial communities in the rhizosphere soil of transgenic maize Ruifeng125 and its non-transgenic control Xianyu 335 at different growth stages is similar, with the largest difference at the ripening stage. The number of bacterial genera in the rhizosphere soil of transgenic maize Ruifeng125 is 5.4% lower than that of the parent control Xianyu 335 at the same period. Therefore, the planting of transgenic maize Ruifeng 125 only affects the diversity and abundance of some bacterial communities in a certain growth period and does not have a significant impact on its rhizosphere soil bacterial community as a whole.

To sum up, the Bt protein secreted by transgenic maize Ruifeng125 into the soil will be degraded in a short time, and thus it will not have a continuous impact on the soil ecosystem. At the same developmental stage, no significant differences in soil bacterial diversity were detected between Ruifeng125 and its non-transgenic control. Compared to the Bt protein, the development stage plays a more important role in the dynamic changes of the bacterial community in rhizosphere soil.

## Figures and Tables

**Figure 1 microorganisms-12-01763-f001:**
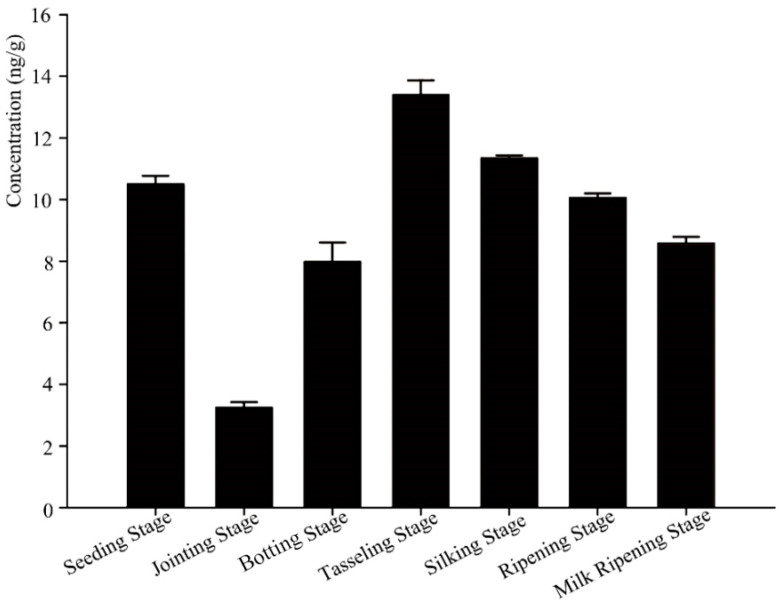
Content levels of Bt proteins in the rhizosphere soil of transgenic maize 125 at different growth stages. Data are presented as the means ± standard deviations of three biological replicates.

**Figure 2 microorganisms-12-01763-f002:**
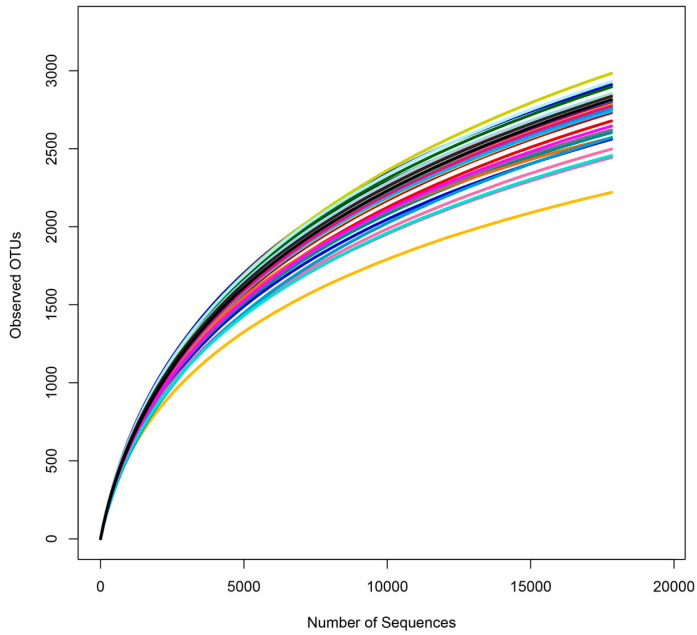
Rarefaction curves of Shannon indices. Different colored curves correspond to different samples.

**Figure 3 microorganisms-12-01763-f003:**
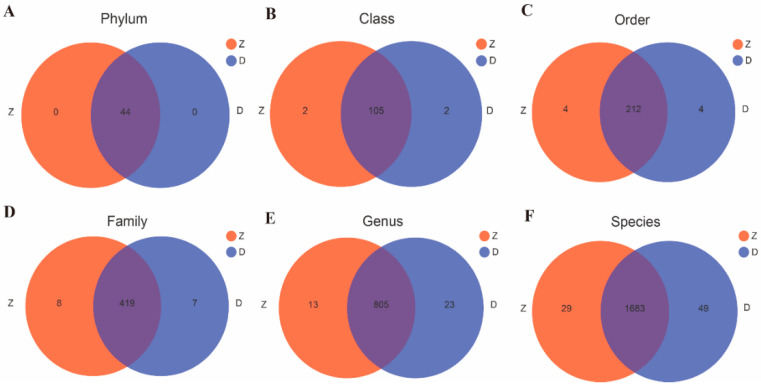
Venn diagram of bacterial community obtained from rhizosphere soil samples of two maize varieties at different classification levels: (**A**) phylum level; (**B**) class level; (**C**) order level; (**D**) family level; (**E**) genus level; (**F**) species level. Z, transgenic maize Ruifeng 125; D, non-transgenic control.

**Figure 4 microorganisms-12-01763-f004:**
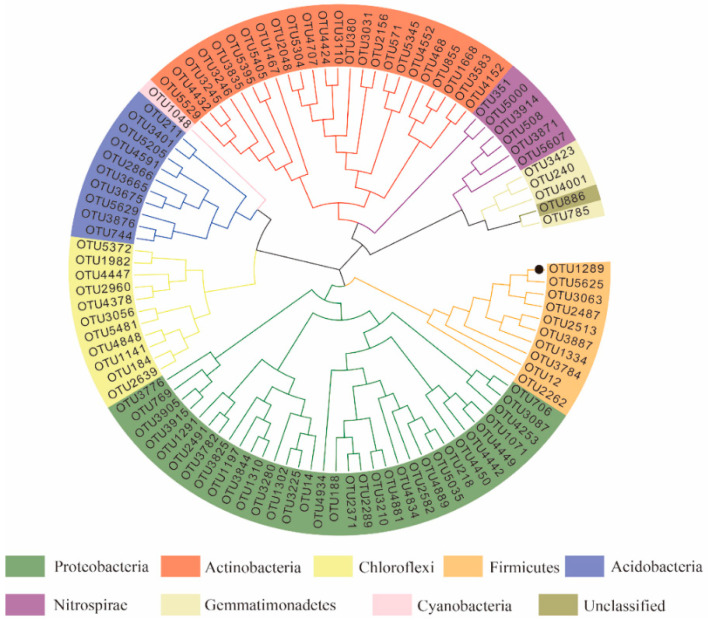
Phylogeny of the bacterial community identified in rhizosphere soil based on the OTU table. The OTU abundance highlighted by the black dot is significantly higher than that of other OTUs.

**Figure 5 microorganisms-12-01763-f005:**
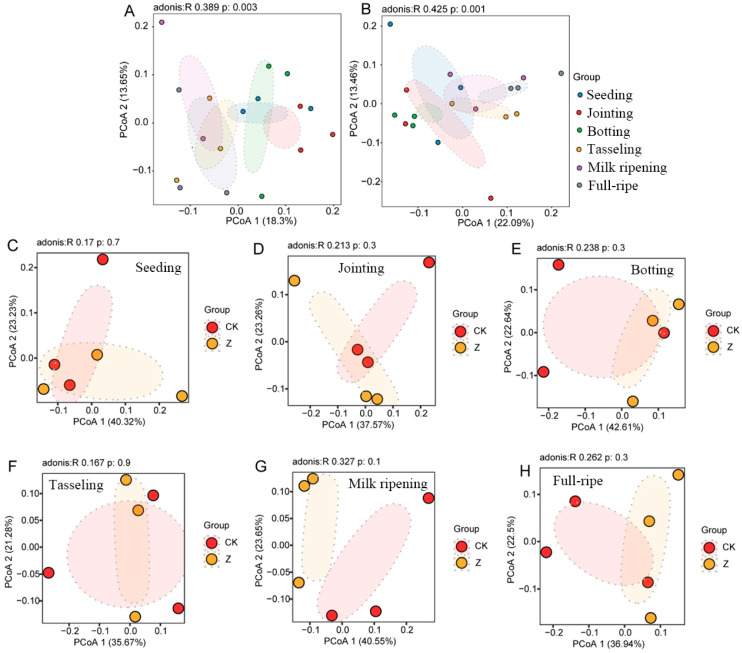
Principal coordinate analysis (PCoA) based on Bray–Curtis distance of the bacterial community in the rhizosphere of transgenic maize Ruifeng125 (**A**) and the control maize variety (**B**) at different growth stages. The *p* value represents the significant difference base on Student’s *t*-test (*p* < 0.05): (**C**) seedling stage; (**D**) jointing stage; (**E**) botting stage; (**F**) tasseling stage; (**G**) milk ripening stage; (**H**) full-ripe stage.

**Figure 6 microorganisms-12-01763-f006:**
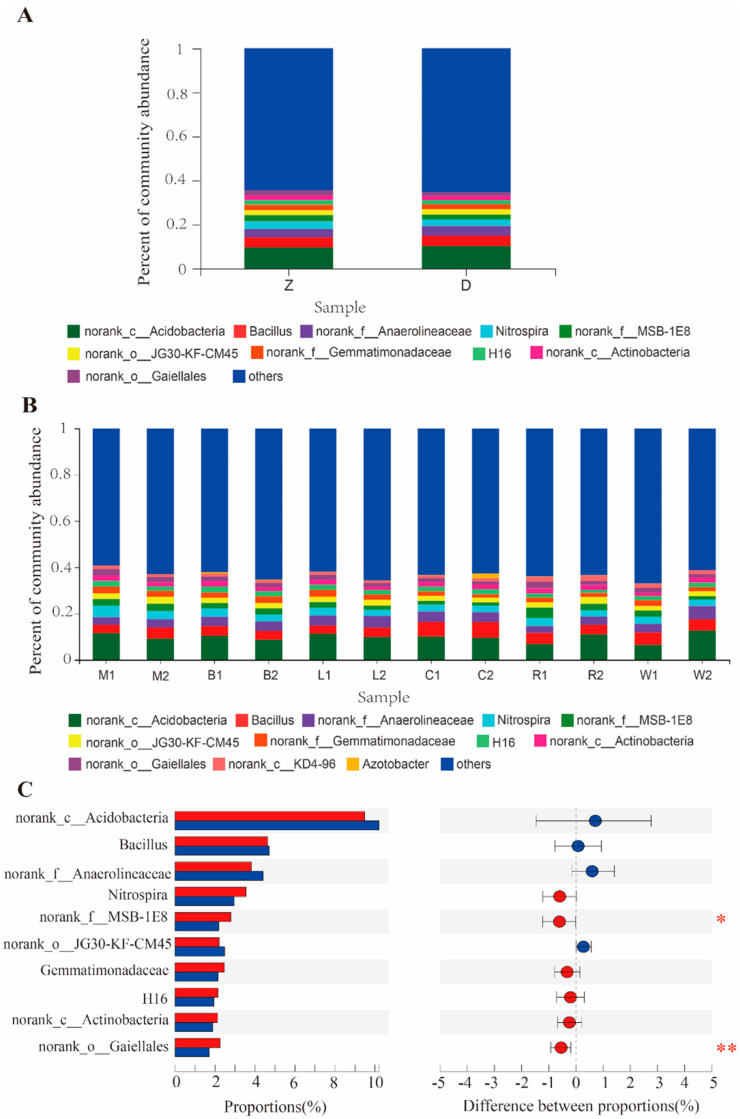
Relative abundance of bacteria in rhizosphere soil of transgenic maize Ruifeng125 and its non-transgenic control. (**A**) Abundance of bacterial community in rhizosphere soil of two kinds of maize at different growth stages. M stands for seedling stage, B stands for jointing stage, L stands for trumpet stage, C stands for tasseling stage, R stands for milking stage, and W stands for ripening stage; 1 represents transgenic maize double resistance 125, 2 represents non-transgenic control Xianyu 335. (**B**) Abundance of bacterial community in rhizosphere soil during the whole life cycle of two kinds of maize. Z represents transgenic maize Ruifeng125, and D represents non-transgenic control Xianyu 335. (**C**) Willcoxon test between two groups. * represents significant difference (*p* < 0.05), ** represents extremely significant difference (*p* < 0.01).

**Figure 7 microorganisms-12-01763-f007:**
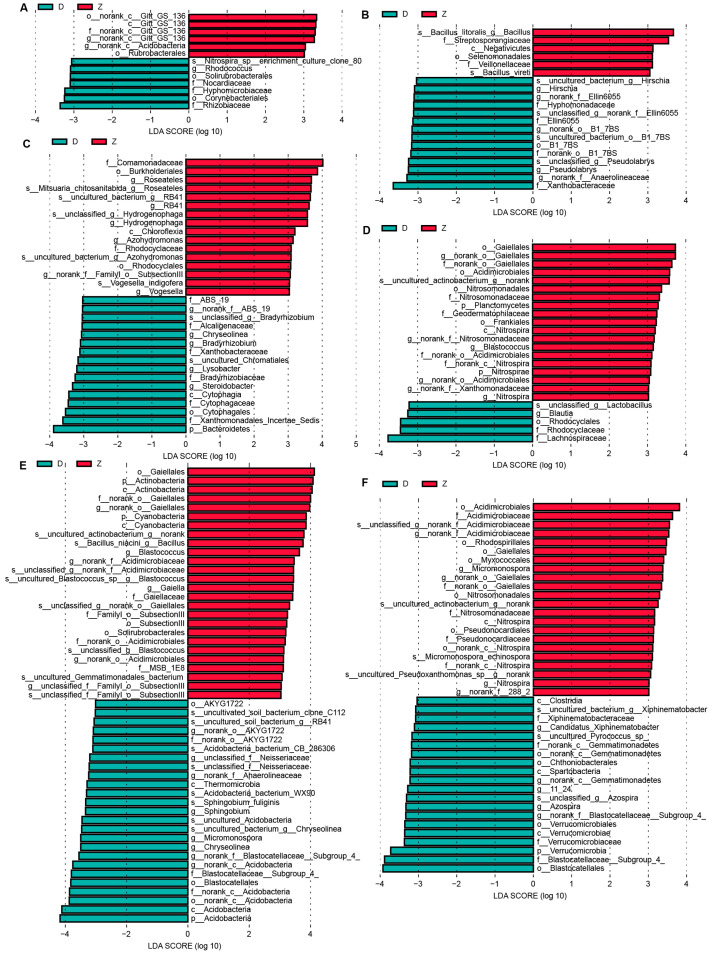
Biomarkers identified by LDA across the 6 growth stages between control and the transgenic maize 125 group: (**A**) seedling stage; (**B**) jointing stage; (**C**) botting stage; (**D**) tasseling stage. (**E**) milk ripening stage; (**F**) full-ripe stage.

**Table 1 microorganisms-12-01763-t001:** The degradation rate of Bt protein after straw was returned to the field.

Degragation (d)	Bt Protein Content (ng/g)	Protein Degradation Rate
0	937.60	0
1	781.68	16.63%
2	549.53	41.39%
3	382.92	59.16%
4	247.71	73.58%
5	152.74	83.71%
6	89.82	90.42%
7	62.26	93.36%
10	25.69	97.26%
20	16.50	98.24%
40	7.69	99.18%
70	1.97	99.79%
110	0.66	99.93%
160	0.35	99.97%
170	--	100%

**Table 2 microorganisms-12-01763-t002:** The number and average length of valid reads and alpha diversity indices observed in two groups.

Group	Valid Reads	Average Length	Shannon Index	Chao Index	ACE Index	Simpson Index	Coverage
Z	38,634 ± 4115	439 ± 0.75	6.99 ± 0.10	4157 ± 289	4160 ± 281	0.0024 ± 0.0003	0.96 ± 0.008
D	38,168 ± 4643	438.934 ± 0.79	7.00 ± 0.14	4051 ± 268	4070 ± 254	0.0025 ± 0.0012	0.96 ± 0.008

## Data Availability

The data presented in this study are available in Zenodo (https://zenodo.org/records/13334964).

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
