# Peer review of "Impact of Transgenic Maize Ruifeng125 on Diversity and Dynamics of Bacterial Community in Rhizosphere Soil"

_microorganisms, 2024, doi:10.3390/microorganisms12091763_

Round 1

Reviewer 1 Report

Comments and Suggestions for Authors

There are some considerations that need to be addressed first;

1. Please follow the journal template, methods are before results.
2. Line 80. Please capitalize, 16s -> 16S.
3. Line 300. add an space, usedto -> uset to.
4. Lines 303 to 305, please review this statement, which is confusing.
5. Line 351, why did you use SILVA123 (from year 2015) as reference database? the newer version is SILVA138.
6. Line 353, I think Wilcoxon test is not appropiate to analyze compositional data, please check for edgeR, ALDEx2, ANCOM or metagenomeSeq to achieve this.
7. Please add to "Data Availability Statement" the BioProject where the raw data is deposited.
8. Lines 107 to 111,Why did you cut sampling depth at 2500?, please add to supplementary materials a table summarizing the reads, raw reads, reads after QC, reads after denoising, reads after removing chimeric sequences, etc.
9. In Table 2, please remove "Average Length" and Chao1 index requires singleton count to be correctly computed, you remove then based on line 346. Please provide the full table, not averaging the groups.
10. In Figure 2, the rarefaction curve of Observed OTU is more informative.
11. Lines 117 to 122, please check the OTU word, it was rewritten by ur text processing software. Also, it is more appropiate to use IndicSpecies package to find specific OTU in the samples.
12. Figure 3, Can you explain how did you compute shared OTU?
13. Did you remove chloroplast and mitochondria sequences from ur OTU table?
14. For the phylogenetic tree, did you perform a bootstrapping to analyze stability?
15. Figure 4, review the caption for the OTU word.
16. Line 143 to Line 145, PCoA is only for visualization, the PERMANOVA implemented as adonis function in vegan is the real test here, rephrase the text explaining it accordingly. Also, you could perform the model including the developmental stage in the model to check the effects for these variables, eg; distances ~ variety*dev_stage. This can help you to support your statement from lines 158-159, also, you wont need 8 plots.
17. Review other methods for differential abundance analysis.
18. Line 180, what does the "_f_" mean?
19. On Figure 6 [A-B], please show the plots using a specific taxonomy level, mixing them makes it confusing. For C, please recompute it with another method.
20. Here you describe LEFSE, it was not described in methods neither cited in the document, please fix it. 
21. On Figure 7, please summarize the results on a specific taxonomy level, or you'll have redundant markers. Adding the abundance information on the markers may be useful.

Author Response

Summary

Thank you very much for your valuable comments and suggestions on our manuscript. The response to your concerns is given point by point in the following and hope that you will find the added information suitable and sufficient for publication.

Comments 1. Please follow the journal template, methods are before results.
Response1: Thank you for pointing this out. The Methods part has been adjusted to the front of Results part.

Comments 2. Line 80. Please capitalize, 16s -> 16S.
Response2: 16s has been changed to 16S.

Comments 3. Line 300. add an space, usedto -> uset to.
Response3: usedto has been changed to used to.

Comments 4. Lines 303 to 305, please review this statement, which is confusing.
Response4: The meaningless text was removed to make the statement clear.

Comments 5. Line 351, why did you use SILVA123 (from year 2015) as reference database? the newer version is SILVA138.
Response5: Sorry for this mistake. SILVA138 was used in this study. The version of SILVA has been corrected.

Comments 6. Line 353, I think Wilcoxon test is not appropiate to analyze compositional data, please check for edgeR, ALDEx2, ANCOM or metagenomeSeq to achieve this.
Response6: Sorry for our fuzzy description of this part. Wilcoxon test was used to compare the difference between paired groups, which has been emphasized in the revised manuscript.

Comments 7. Please add to "Data Availability Statement" the BioProject where the raw data is deposited.
Response7: All the data of this study have been deposited in Zenodo which served as a developed tools for Big Data management and extended Digital Library capabilities for Open Data. The Download link of the raw data has been added to the Data Availability Statement part.

Comments 8. Lines 107 to 111,Why did you cut sampling depth at 2500?, please add to supplementary materials a table summarizing the reads, raw reads, reads after QC, reads after denoising, reads after removing chimeric sequences, etc.
Response8: Sorry for our unclear description about the sampling depth at 2500. As figure 2 shown, initially, as the depth of sequencing increased, the diversity of species increased dramatically. The Shannon's dilution curve tends to be gently when the sampling depth is more than 2500. The description was re-organized in the revised manuscript.

Comments 9. In Table 2, please remove "Average Length" and Chao1 index requires singleton count to be correctly computed, you remove then based on line 346. Please provide the full table, not averaging the groups.
Response9: Full table 2 has been provided in the revised manuscript.

Comments 10. In Figure 2, the rarefaction curve of Observed OTU is more informative.
Response10: The rarefaction curve of Observed OTU has been added as Figure 2.

Comments 11. Lines 117 to 122, please check the OTU word, it was rewritten by ur text processing software. Also, it is more appropiate to use IndicSpecies package to find specific OTU in the samples.
Response11: We have checked the word, and no errors were found.

Comments 12. Figure 3, Can you explain how did you compute shared OTU?
Response12: We computed the special and shared OTUs mainly according to the OTU table. If the average abundance of an OTU was not zero, we assumed that the group owned the OUT.

Comments 13. Did you remove chloroplast and mitochondria sequences from ur OTU table?
Response13: Yes, the hloroplast and mitochondria sequences have been removed in the steps of quality control.

Comments 14. For the phylogenetic tree, did you perform a bootstrapping to analyze stability?
Response14: The phylogenetic tree was structured after 1000 times bootstrapping, which was marked in the revised manuscript.

Comments 15. Figure 4, review the caption for the OTU word.
Response15: Mistakes of the caption for OTU has been corrected.

Comments 16. Line 143 to Line 145, PCoA is only for visualization, the PERMANOVA implemented as adonis function in vegan is the real test here, rephrase the text explaining it accordingly. Also, you could perform the model including the developmental stage in the model to check the effects for these variables, eg; distances ~ variety*dev_stage. This can help you to support your statement from lines 158-159, also, you wont need 8 plots.
Response16: We agree with your suggestions that the PERMANOVA implemented as adonis function in vegan is the real test. The text of the PERMANOVA results have been added to this part.

Comments 17. Review other methods for differential abundance analysis.
Response17: Specific methods for differential abundance analysis has been adjusted and marked in Methods part.

Comments 18. Line 180, what does the "_f_" mean?
Response18: Sorry for this mistake. The _f_ have been replaced by the OUT belong to “Gemmatimonadaceae”.

Comments 19. On Figure 6 [A-B], please show the plots using a specific taxonomy level, mixing them makes it confusing. For C, please recompute it with another method.
Response19: Figure 6 [A-B] showed the plots at the specific genus level like Bacillus, Nitrospira and H16. However, some genus were not classified to a specific genus, and were marked by norank and followed by the family name. For C, the method for differential analysis were replaced and marked in the legend.

Comments 20. Here you describe LEFSE, it was not described in methods neither cited in the document, please fix it. 
Response20:  The description of LEFSE has been added to the methods.

Comments 21. On Figure 7, please summarize the results on a specific taxonomy level, or you'll have redundant markers. Adding the abundance information on the markers may be useful.

Response21: We appreciate your comment. The results of LEFSE have been summarized at genus or family levels.

Reviewer 2 Report

Comments and Suggestions for Authors

In the peer-reviewed manuscript Authors used NGS technology to establish if genetically modified maize in comparison to non-modified variety differently influence the population of bacteria in rhizosphere.  This study is among other similar ones to answere the question to be about the safety use of such crops. According to conclusion drawn by Authors only minor changes in soil bacteria composition occured when maize Ruifeng125 was apllied.

General comment

Authors made annotation one OTU per species what can generate  inaccurate conclusions. Such approach do not cover discrete differences inside the species i.a. pathogenic versus non-pathogenic phenotypes. It would be better make annotation - one OTU per phenotype. Below a quote from https://drive5.com/usearch/manual/otus.html

„With the UNOISE algorithm the goal is to report all correct biological sequences in the reads. These are called zero-radius OTUs, or ZOTUs. It is expected that some species may be split over several ZOTUs due to intra-species variation (differences between paralogs and differences between strains). The advantage of ZOTUs is that they enable resolution of closely related strains with potentially different phenotypes that would be lumped into the same cluster by 97% OTUs”.

Therefore, the data obtained in current study are not proof thus it should be confirmed in biological studies in future.

Detailed comments:

In the introduction chapter I would recommend Authors shortly explain the function of Bt protein and mechanism of action.

l. 91, 209, 214, 232, 279 ; in this lines different description of transgenic maize „12-5” instead ‘125’

l.93 please rephrase the sentence to make it better understood

l.129 There is no information about what this data concerns, transgenic corn, unmodified corn or both.

l. 137 copolymer?

l. 140 based on which Table? And OTUs and not OTU

l. 141 should be ‘dot’ not ‘dots’

Figure 5 A,B please give the groups not in alphabetical order but in growth phase order; additionally, the use of the letter B for jointing stage causes confusion when reading the figure caption, where (B) is for the graph and the stage of corn development, I propose to change it; for graphs C-H I would propose to use the same colors as in graphs A,B for particular samples; it would be more friednly for readers; Group captions on graphs C-H are confusing, the lack of explanation

l. 179 after the word ‘norank’ should be the name and the end of sentence is „_f_”; Did you ment norank_f_MSB_1E8?

l. 180 in sentence is „Gaillares” and I think it should be Gaiellales

Figure 6. In my opinion, graphic B should be before A. i.e. in reverse order

l. 219 – 226. Are you sure that Rhodococcus and not Nitrospira? for (D) in Fig.A; Pseudolabrys and not Ellin6055 for (D) in Fig.B; Steroidobacter and not Lysobacter for (D) in Fig. C; and Blautia and not Lactobacillus for (D) in Fig. D; PrzywoÅ‚aj w tekÅ›cie l. 226 (Figure 6D).

l.222 please, keep the same order as befor what means first data for control, and then for transgenic maize since it is difficult to follow by; moreover, please list species in ascending order of abundance

Figure 7 Unreadable figure

l. 279 also different name of the transgenic maize variety exist in the text „Shuangkang” instead „Ruifeng125”

l. 282 This citation [27] is incorrect. This publication study the influence of different factors on the Bt peptide degradation and not on the rhizosphere soil bacterial community

l. 298 What kind of fertilization? It is known that some minerals can influence the population of rhizopheric bacteria more then plant exudates. Please discuss it also in the end chapter.

l. 304 I think that instead „Riley, D.; Barber, S.D. Salt accumulation at the soybean (Glycine max (L.) Merr.) 304 root-soil interface. Soil Sci. Soc. Am. J. 1970, 34) „ should be number of ref. [] which is not included in the list of references.

l. 326 The Authors should include in the description what was the positive control (Cry1Ab/Cry1Ac included in the kit)

l. 327-331 please rephrase the sentence to make it better understood

Comments on the Quality of English Language

Comments on language corrections included in main review. Overall, language is understandable.

Author Response

Summary

Thank you very much for your valuable comments and suggestions on our manuscript. The response to your concerns is given point by point in the following and hope that you will find the added information suitable and sufficient for publication.

Comments 1:In the introduction chapter I would recommend Authors shortly explain the function of Bt protein and mechanism of action.

Response1: We appreciate your comment. The function of Bt protein and mechanism of action have been added to the introduction.

Comments 2:l. 91, 209, 214, 232, 279 ; in this lines different description of transgenic maize „12-5” instead ‘125’

Response2: “12-5” has been instead of “125” throughout the manuscript.

Comments 3:l.93 please rephrase the sentence to make it better understood

Response3: The sentence has been rephrased.

Comments 4:l.129 There is no information about what this data concerns, transgenic corn, unmodified corn or both.

Response4: The sentence has been rephrased.

Comments 5:l. 137 copolymer?

Response5: The puzzled word ‘copolymer’ has been removed.

Comments 6:l. 140 based on which Table? And OTUs and not OUT

Response6: All the OUT have been replaced by OTU throughout the manuscript.

Comments 7:l. 141 should be ‘dot’ not ‘dots’

Response7: “dots” has been instead of “dot”

Comments 8:Figure 5 A,B please give the groups not in alphabetical order but in growth phase order; additionally, the use of the letter B for jointing stage causes confusion when reading the figure caption, where (B) is for the graph and the stage of corn development, I propose to change it; for graphs C-H I would propose to use the same colors as in graphs A,B for particular samples; it would be more friednly for readers; Group captions on graphs C-H are confusing, the lack of explanation

Response8: Figure 5 has been modified according to your suggestion to making it clearer.

Comments 9:l. 179 after the word ‘norank’ should be the name and the end of sentence is „_f_”; Did you ment norank_f_MSB_1E8?

Response9: The word ‘norank’ was meaningless here which has been stead of the specific name of “Gemmatimonadaceae”.

Comments 10:l. 180 in sentence is „Gaillares” and I think it should be Gaiellales

Response10: ‘Gaillares’ has been stead of ‘Gaiellales’.

Comments 11:Figure 6. In my opinion, graphic B should be before A. i.e. in reverse order

Response11: We agree with your suggestion. Figure 6B has been adjusted to be before A.

Comments 12:l. 219 – 226. Are you sure that Rhodococcus and not Nitrospira? for (D) in Fig.A; Pseudolabrys and not Ellin6055 for (D) in Fig.B; Steroidobacter and not Lysobacter for (D) in Fig. C; and Blautia and not Lactobacillus for (D) in Fig. D; PrzywoÅ‚aj w tekÅ›cie l. 226 (Figure 6D).

Response12: Here, we performed the LEFSE analysis and focused the core taxa at the genus level. Therefore, only the genus (LDR>3) was mentioned in this part.

Comments 13:l.222 please, keep the same order as befor what means first data for control, and then for transgenic maize since it is difficult to follow by; moreover, please list species in ascending order of abundance

Response13: Thanks for your suggestion, the order of abundance as been adjusted.

Comments 14:Figure 7 Unreadable figure

Response14: Figure 7 has been modified to keep readable.

Comments 15:l. 279 also different name of the transgenic maize variety exist in the text „Shuangkang” instead „Ruifeng125”

Response15: “Shuangkang” has been instead wit “Ruifeng

Comments 16:l. 282 This citation [27] is incorrect. This publication study the influence of different factors on the Bt peptide degradation and not on the rhizosphere soil bacterial community

Response16: Sorry for this mistake, the citation here were meaningless and has been removed.

Comments 17:l. 298 What kind of fertilization? It is known that some minerals can influence the population of rhizopheric bacteria more then plant exudates. Please discuss it also in the end chapter.

Response17: We rechecked all the experimental records and found no fertilization treatment was performed. The description has been rephased here.

Comments 18:l. 304 I think that instead „Riley, D.; Barber, S.D. Salt accumulation at the soybean (Glycine max (L.) Merr.) 304 root-soil interface. Soil Sci. Soc. Am. J. 1970, 34) „ should be number of ref. [] which is not included in the list of references.

Response18: Sorry for this mistake. Just as you think, this disorder text was the result of our misquoting. Now, the right reference has been added to this place.

Comments 19:l. 326 The Authors should include in the description what was the positive control (Cry1Ab/Cry1Ac included in the kit)

Response19: The positive control is the Cry1Ab/Cry1Ac protein with the purity more than 95%. We have added more description to this part.

Comments 20:l. 327-331 please rephrase the sentence to make it better understood

 Response20: This sentence has been rephrased.

Round 2

Reviewer 1 Report

Comments and Suggestions for Authors

The manuscript presentation has been improved